# Single CsPbBr_3_ Perovskite Microcrystals: From Microcubes to Microrods with Improved Crystallinity and Green Emission

**DOI:** 10.3390/ma17164043

**Published:** 2024-08-14

**Authors:** Khouloud Abiedh, Marco Salerno, Fredj Hassen, Zouhour Zaaboub

**Affiliations:** 1Micro-Optoelectronics and Nanostructures Laboratory (LR99/ES29), Faculty of Sciences, University of Monastir, Monastir 5000, Tunisia; abiedh.khouloud@fsm.u-monastir.tn (K.A.); fredj.hassen@fsm.u-monastir.tn (F.H.); zouhour.zaaboub@isimm.rnu.tn (Z.Z.); 2Department of Physics, Institute for Globally Distributed Open Research and Education (IGDORE), University of Genoa, Via Dodecaneso 33, 16146 Genoa, Italy

**Keywords:** perovskite, microrods, aminosilane, optical properties, surface modification

## Abstract

All-inorganic perovskite materials are promising in optoelectronics, but their morphology is a key parameter for achieving high device efficiency. We prepared CsPbBr_3_ perovskite microcrystals with different shapes grown directly on planar substrate by conventional drop casting. We observed the formation of CsPbBr_3_ microcubes on bare indium tin oxide (ITO)-coated glass. Interestingly, with the same technique, CsPbBr_3_ microrods were obtained on (3-Aminopropyl) triethoxysilane (APTES)-modified ITO-glass, which we ascribe to the modification of formation kinetics. The obtained microcrystals exhibit an orthorhombic structure. A green photoluminescence (PL) emission is revealed from the CsPbBr_3_ microrods. Contact angle measurements, Fourier-transform infrared and PL spectroscopies confirmed that APTES linked successfully to the ITO-glass substrate. We propose a qualitative mechanism to explain the anisotropic growth. The microrods exhibited improved PL and a slower PL lifetime compared to the microcubes, likely due to the diminished occurrence of defects. This work demonstrates the importance of the substrate surface to control the growth of perovskite single crystals and to boost the radiative recombination in view of high-performance optoelectronic devices.

## 1. Introduction

Cesium halide perovskites CsPbX_3_ (X=Cl, Br and I), which are all-inorganic, have recently attracted high interest in optoelectronic applications [1,2] thanks to their solution processability [3], low cost [4], high stability [5] and narrow emission line [1]. These members of the perovskite family appear to be a promising alternative to traditional semiconductors, but require improvement in the synthesis and understanding of their properties. Several synthesis protocols have been developed in order to obtain all-inorganic perovskite in different forms, such as microcrystals [6], thin films [1,7] and nanocrystals (NCs) [4]. However, most of those strategies do not allow for the proper control of shape and size. The synthesis of material with high quality and well-defined morphology not only benefits fundamental research, but also offers great promise for practical applications [8]. In the arena of NCs, synthesis protocols based on the solution phase, such as hot injection [9] or ligand assisted precipitation [10], have shown their potential in regulating the final morphology of the CsPbBr_3_ NCs and promoting the selective formation of nanocubes, nanoplates and nanowires by acting on the acid–base equilibrium mechanism [11]. NCs with a variety of shapes have been synthesized by kinetically controlling the growth of particular facets [12] or through an oriented attachment mechanism [13]. The surface chemistry of NCs is decisive in determining the final morphology of the material [14]. Surfactant or surface ligands such as alkylamine and carboxylic acid have been proven essential to alter the kinetic pathways and give anisotropically developed nanostructures, forming hydrogen bonds with bromide anionic surface sites and alkyl carboxylate binding the surface cationic sites [12]. However, the shape control of single microcrystals is less explored.

In this study, we investigated the effects of the (3-aminopropyl)triethoxysilane (APTES) surface ligand-modified substrate on the growth of CsPbBr_3_ perovskite single microcrystals. A silane layer is usually employed to control the physical and chemical properties of a solid surface [11]. APTES is a surface ligand extensively employed to functionalize surfaces with amino silane via covalent interaction [15]. The attachment of primary amine seems to be an effective technique to control the growth of all inorganic CsPbBr_3_ perovskites. Our work shows an unprecedented method to prepare CsPbBr_3_ single microcrystals via a solution processing strategy, which gave rise to the uniaxial directional growth of elongated crystals that we called microrods (MRs) crystallizing in the orthorhombic phase. The presence of functional groups of silane layers and the possible chemical bonding to CsPbBr_3_ perovskite was confirmed by Fourier-transform infrared spectroscopy (FTIR). The synthesis technique in this study is facile and offers a new procedure to grow CsPbBr_3_ perovskite MR structures with improved properties for optoelectronic application.

## 2. Materials and Methods

### 2.1. Surface Treatment

If not otherwise specified, all chemicals and materials were acquired from Sigma-Aldrich (St. Louis, MO, USA). ITO-coated glass wafers to be used as substrates were rinsed sequentially with absolute ethanol and acetone in an ultra-sonicator bath. The cleaned wafers were hydroxylated via Piranha treatment at room temperature (RT) for 20 min and with H_2_O_2_/H_2_SO_4_ (3:7) at 60 °C, then rinsed with water and dried with argon flux. For the silanization process, the substrates were immersed in 2% toluene solution for 3 min. Finally, the substrates were rinsed with toluene and deionized water and then dried in a nitrogen stream.

### 2.2. Perovskite Preparation

We prepared the CsPbBr_3_ precursor solution by mixing CsBr and PbBr_2_ in dimethyl sulfoxide (DMSO). The solution of the precursor, at a 0.3 M concentration, was magnetically stirred before deposition. A single drop of 40 µL volume was deposited on the silane-functionalized surface by drop casting. After depositing onto the substrate, annealing was carried out at 180 °C for 20 min.

### 2.3. Characterization

X-ray diffraction (XRD) patterns were obtained using an Empyrean X-ray diffractometer (Malvern Pananalytical, Malvern, UK) equipped with a 1.8 Kw CuKα ceramic X-ray tube and operating at 45 kV and 40 mA.

Optical microscopy images and 3D profilometry measurements were obtained with a confocal Zeta-20 profilometer (Zeta instruments, Milpitas, CA, USA).

Atomic force microscopy (AFM) measurements were performed with a XE 100 (Park Systems, Suwon, Republic of Korea) microscope under ambient atmosphere.

PL microscopy was used to obtain images and to acquire the spectrum of individual microcrystals under 488 nm wavelength (blue light) excitation, with collection in the cyan-green window (500–550 nm).

UV–Visible absorbance spectra were measured using a SPECORD 210-PLUS spectrophotometer equipped with a deuterium D2E lamp (185–350 nm) and a halogen lamp (330–1100 nm).

Steady state PL spectra were measured at 300 K using a laser excitation wavelength of 360 nm. The spectral analysis of the luminescence measurements was dispersed using a iHR320 monochromator (HORIBA Jobin Yvon, Kyoto, Japan) and detected by a silicon photodiode.

Time-resolved PL (TRPL) measurements of the recombination dynamics were carried out by a time-correlated single photon counting setup based on a pulsed laser diode ($\lambda_{ex}=375\;\mathrm{nm})$ operated at 80 MHz and a time Harp 200 electronic card from Pico Quant company [16]. The excitation laser pulse width was 70 ps, and the time resolution of the experimental setup was 40 ps. The lifetimes *t* were obtained by fitting the experimental data of PL intensity *I_PL_*, as compared to initial intensity *I*_0_, to a mono-exponential decay:(1)$$I_{PL}\left(t\right)=I_{0}\;e^{-\frac{t}{\tau }}$$

Contact angle (CA) measurements were carried out by the CA goniometer Digidrop GBX using the Laplace-Young method [17].

FTIR spectra were recorded using a Specord 210 plus spectrometer (Analytik Jena, Jena, Germany) equipped with a universal attenuated total reflectance mode and scanned at RT in transmission mode over the wavenumber range of 4000–850 cm^−1^.

## 3. Results and Discussion

CsPbBr_3_ microcrystals were prepared on solid substrates in two forms: the novel MRs (elongated in one planar direction) and the planar-symmetric microcrystals already presented in previous works [18,19], which we call here microcubes (MCs). Actually, the name MCs is to simplify the description, as the microcrystals are not full cubes, but have a hollow pyramidal pit on top, discussed extensively in our previous reports [18,19]. The MCs will be used here as a term of comparison for the novel MR structures. A morphological characterization of the MCs is reported in Figure 1a,b,e. An optical micrograph of the sample (Figure 1a) shows the presence of square-shaped structures (see also the 3D optical profilometer image of the same region in Figure 1b). The MCs are quite homogenous in size (see distribution in Figure 1e) with an edge length of ~64 µm. The preparation parameters are critical for the final microstructure size. The XRD pattern (Figure 1h) evidences the crystalline nature of the MCs, which matches the orthorhombic perovskite phase. The UV–visible absorption spectrum of the as-prepared sample presented in Figure 1j shows the typical profile of CsPbBr_3_ microcrystals, characterized by an absorption peak at ~517 nm. The PL spectrum of the MCs shows a narrow and intense band (fullwidth at half maximum ~17 nm) centered at ~519 nm (2.39 eV). The emission band is associated to the bulk bandgap recombination of CsPbBr_3_ [19].

Figure 1c,d present some perovskite MRs deposited on APTES-modified ITO-glass substrate. The images show the formation of well-defined rectangular single rods. Obviously, the effect of the APTES-modified substrate was the transformation of crystals from MCs to MRs. In the given fabrication conditions, the aspect ratio of the MRs (ratio of long side to short side) is ~5, while the longest diagonal is ~200 µm. The size distribution histogram for the latter parameter is presented in Figure 1f, whereas Figure 1g shows the size distribution of the short MR side. According to the histogram in Figure 1f, the spread around the peak size value is described by a standard deviation of ~30 µm. For the off-plane size of the MRs, optical profilometer measurements (see a representative 3D image in Figure 1d) showed a MR thickness in the range of 10 to 25 µm.

The bulk crystallinity of both MCs and MRs was confirmed by XRD measurements (see Figure 1h,i, respectively). The majority of peaks, the green dots in the diffractogram, fits well with the CsPbBr_3_ orthorhombic perovskite phase (PDF#45-0752, standard diffraction pattern), confirming the formation of CsPbBr_3_ crystals. The sharp peaks indicate the high crystallinity of the MRs. The XRD pattern revealed also the presence of other phases such as Pb (cubic, F m-3 m) and BrN (tetragonal, P 4/n m m).

High-resolution confocal microscopy images (2D and 3D) of the novel MRs have been acquired (Figure 2a,b). Clearly, one can see that the MRs fluoresce in the green region of the visible electromagnetic spectrum. Two PL spectra extracted from different regions of the rods in the confocal image (red and green circles in Figure 2b) are presented in Figure 2c. It appears that the edge of the MR yields a higher PL as compared to the center and a peak shift towards the red (~530 nm vs. ~520 nm). This effect has been confirmed by testing several different MRs. We should point out that our MRs present a hollow top with an inverted pyramid shape similar to the MCs (see Figure 1b,e). We have shown previously by using PL microscopy that in single CsPbBr_3_ MCs [18], the emission from the upper surface (near the edge) is different compared to the one from the center. The PL intensity changes by changing the excitation surface position for single CsPbBr_3_ microcrystal.

For a deeper understanding of the optical properties of the MRs, a more advanced optical characterization has been carried out. In Figure 2d,e, the typical absorbance and steady state PL spectrum of the MRs recorded at 300 K are presented, respectively. As we can see, the PL spectrum revealed the presence of two peaks. One peak, named $P_{1}$, is located at 442 nm and can be assigned to APTES emission [20]. The important emission peak that presents an asymmetrical profile with an extended shoulder on the long wavelength side was fitted by two Gaussians with maxima located at 523 and 540 nm, which were denoted by *P*_2_ and *P*_3_, respectively. This feature can be attributed to the PL from the CsPbBr_3_ MRs. Correspondingly, a strong absorbance band at ~520 nm is shown in Figure 2d, which is a characteristic peak of bulk CsPbBr_3_ perovskite [21]. Figure 2f displays normalized TRPL emission profiles recorded for the sample at RT at different energies of the emission peaks. It can be seen that the profile of $P_{1}$ decays faster than that of $P_{2}$. The PL decays presented the following lifetimes: t_1_~1.62 ns, t_2_~4.12 ns and t_3_~7.86 ns for peaks $P_{1}$, $P_{2}$ and $P_{3}$, respectively. Obviously, the longer carrier lifetime results from the delayed recombination process via non-radiative pathways, which means that the $P_{3}$ PL peak can be assigned to the emission of shallow defects in the CsPbrBr_3_ material [21].

CA measurements were carried out to determine the wettability of substrates before and after APTES modification, which are shown in Figure 3a. The obtained CA value for bare ITO-glass was found to be ~39° (Figure 3(a1)). A significant increase in CA, ~64° (Figure 3(a2)), was observed for the APTES-modified surface (Figure 3b,c). Thus, the surface became less hydrophilic upon coating with APTES due to the presence of the hydrophobic alkyl chains of this molecule [22,23] (see Figure 3b). The APTES is anchoring the ITO-glass surface by replacing the terminating OH groups of the substrate with organic groups attached to amine groups (see Figure 3c), which changes the surface energy and thus its wettability.

Typically, the silanization process consists of two steps: hydrolysis and condensation. In the first step, the hydrolysis of the substrate leads to the formation of the reactive silanol group (R-Si-OH) and ethanol molecules [22]. In the second step, the condensation of these silanol groups takes place, leading to the formation of siloxane (Si-O-Si) linkages over the surface [22,23]. Thus, the attachment of APTES takes place with OH groups on the substrate, and a NH_2_ terminated layer is created on the surface (Figure 3c). These amino (-NH_2_) groups on the APTES surface react with perovskite via chemical bonding. According to the literature [24,25], the attachment of APTES to CsPbBr_3_ perovskite takes place through three possible binding mechanisms [15]: (1) the formation of dative-covalent bonds between amines (NH_2_) of silanes network and undercoordinated Pb^2+^ sites; (2) hydrogen bond formation between the remaining silanol and surface bromide (Br^−^) sites; and (3) the formation of hydrogen bonds through protonated amines (NH^+^) and bromine (Br^−^) dangling bonds.

For ideal silanization, all amino groups in the APTES should be oriented away from the substrate surface [26]. The typical APTES surface quality was as shown in the representative image in Figure 3d, showing a flat and smooth background with some aggregates up to ~1 mm in size, likely due to excess material. In fact, the thickness of the APTES layer was also estimated by using AFM across hand-made scratches in the molecular coating, which resulted in a value around ~150 nm. The thickness of one silane layer is ~2 nm [27], which confirms the formation of multisilane layers on the top of ITO-glass substrate.

The surface chemical composition of CsPbBr_3_ MRs was characterized by FTIR measurements (see black line in Figure 3e). The resulting peaks assignment is reported in Table 1. FTIR results confirm the presence of the asymmetric stretching vibration of surface hydroxyl groups (OH) by the broad band at 3464 cm^−1^ due to ethanol. This large band shows a small peak located at 3258 cm^−1^ due to the N-H symmetric stretching vibration of APTES [28]. Bands at 2856 cm^−1^, 2925 cm^−1^ and 2959 cm^−1^ correspond to C-H vibration from both ethanol and APTES [28]. Bands at 1638–1664 cm^−1^ are attributed to the binding vibration of primary amines (NH_2_) in APTES [28]. The appearance of these bands demonstrated that amine functional groups in organosilane were successfully grafted onto the ITO-glass substrate [28]. Furthermore, peaks corresponding to Si-O-Si bonds were observed at around 1012 cm^−1^–1066 cm^−1^, indicating the condensation reaction between silanol groups [28]. These bands confirm the condensation reaction between the ethoxy groups of APTES and the ITO surface hydroxyl groups, namely ITO-glass modification with APTES. The FTIR spectrum shows another weak peak at 1575 cm^−1^, which was assigned to the NH^+^ bond [9].

For comparison, in Figure 3e, the FTIR spectrum of the MCs sample has also been reported (red line). There, no intense peaks can be seen because CsPbBr_3_ is an inorganic compound. However, a weak peak arising from the interaction between DMSO and Pb^2+^ appears at 1053 cm^−1^, ascribed to the stretching vibration of S=O, confirming the formation of a CsBr⋅PbBr_2_⋅DMSO complex [29]. The peak at 755 nm-896 nm can be assigned to oxygen stretching vibration, as the measurement was conducted in ambient conditions.

In conclusion, FTIR spectroscopy confirms the presence of the characteristic functions of APTES on the sample surface and the attachment of perovskite to the APTES through three different binding due to the presence of NH_2_, NH^+^ and silanol groups on the surface of the rods (see Figure 3f).

To figure out the origin of the anisotropic growth of CsPbBr_3_ MRs, we propose a plausible mechanism. In a precursor solution, the Br^−^ ions coordinate with Pb^2+^ to form [PbBr_6_]^4−^ octahedra in DMSO. After precursor solution deposition, the solvent is evaporated under heat treatment to allow for the crystallization of perovskite [30], and the Cs^+^ ions occupy the octahedral sites to form CsPbBr_3_ perovskite MCs above the bare ITO-glass substrate. This occurs by using the same precursor solution for both MRs and MCs, which means that MRs evolve from MCs during the heating process. The crystallization process (nucleation and growth) plays an important role in determining the crystal structure, shape and size [31]. According to the literature, nano/micro or macro crystal synthesis with various sizes and shapes can be described via the supersaturation chemistry approach (Lamer theory) [31,32]. The shape control of CsPbX_3_ perovskite NCs has been tuned either by kinetically controlling the growth of particular crystal facets [12] or through an oriented attachment mechanism [13]. In contrast, few studies deal with the anisotropic microparticles formation mechanism. Consequently, a hypothetical approach to understanding the mechanism of microcrystal construction provides a greater control over the size, shape and structure of single crystals, and consequently, the ability to tune the properties by varying the crystallization conditions.

The growth rate of a single crystal increases with the increase in CA. The less hydrophilic APTES-modified surface causes precursor ions and solvent to become closer, which will increase the diffusion of ions, leading to faster crystal growth. In fact, it is known that at high growth rates, organic molecules (ligands) that selectively adhere to particular NC facets decrease their energy and slow down their growth [33]. High-energy facets grow faster than low-energy ones, giving rise to the formation of anisotropic shapes [34]. Previously, Pan et al. [9] suggested the preferential binding of amine toward certain facets during the growth of CsPbBr_3_ nanowires and nanoplatelets. Also, it is demonstrated that ammonium binding with Cs^+^ ions on the surfaces allows the growth of platelets [34]. Therefore, we suppose here that the amine groups compete with the Cs^+^ ions on the surface of the growing rods, and thus, they slow the growth along the orthogonal direction, leading to the observed anisotropic rod structures. Also, it is suggested that a halide-rich environment may result in oriented crystal growth along special facets.

In support of our hypothesis that the preferential binding of APTES takes place through amine groups, giving rise to the formation of perovskite MRs, we have compared the PL properties of MCs and MRs. The PL spectrum of MRs is slightly redshifted with respect to that of the MCs (see Figure 4a). The PL maximum slightly redshifts from 519 nm for MCs to 523 nm for MRs. The MRs also show PL enhancement compared to the MCs. The origin of enhancement may arise from effective surface passivation by APTES molecules. TRPL spectra of the microcrystals were also acquired and fitted using a biexponential decay function where *A_i_* and τ*_i_* refer to the decay amplitudes and respective PL lifetimes for *i* = 1,2. The average lifetime can thus be obtained by the following formula:(2)$$\tau_{av}=A_{1}\tau_{1}+A_{2}\tau_{2}$$

The time-resolved measurements also show an enhancement of the PL lifetime upon APTES treatment (see Figure 4b) from ~2.3 to ~7.4 ns (details see Table 2), whereas, from the XRD patterns of Figure 1g,h, one can see that the MCs have narrower FWHM and stronger crystallization than the MRs and the PL lifetime of MCs is shorter than that of MRs. The reason is probably because, thanks to the crystal adhesion being improved by APTES, the overall coverage of the APTES-treated substrate by CsPbBr_3_ microcrystals is higher, with small anisotropic shape microcrystals appearing between the MRs (confocal 3D imaging). When no APTES is present and MCs form, the empty spaces between the microcrystals increase the nonradiative recombination rate and therefore decrease the PL lifetime. Additionally, the crystallites appearing between the MRs are smaller, and this decrease in size causes the XRD peak to broaden.

## 4. Conclusions

In summary, we presented a comparative study of CsPbBr_3_ perovskite microcrystals with different forms grown directly on a planar substrate by the conventional drop casting method. We observed the formation of CsPbBr_3_ MCs on a bare ITO-coated glass substrate. With the same technique, CsPbBr_3_ single MRs were obtained on an APTES surface ligand-modified ITO-glass substrate, which we ascribed to the modification of the kinetic regime by using APTES. Green emission was revealed from the CsPbBr_3_ microcrystals by using PL spectroscopy. CA measurements, FTIR and PL spectroscopies confirmed that APTES linked successfully to the ITO-coated glass substrate. The fabricated perovskite MRs exhibited improved PL thanks to the introduced APTES molecules effectively passivating surface defects and controlling crystal growth. We conclude that, even for microcrystals, the introduction of certain organic molecules that selectively adhere to a particular crystal facet can be used to slow the growth of that side relative to others, leading to the formation of various shapes in single crystals. Thus, we have identified a new strategy to enhance the photoluminescence of and to control the shape of CsPbBr_3_ microcrystals for optoelectronic applications. Further research work is needed to confirm these findings. Still, our work points out that the careful management of perovskite MCs and substrate surface engineering are important factors in this promising class of optoelectronic materials.

## Figures and Tables

**Figure 1 materials-17-04043-f001:**
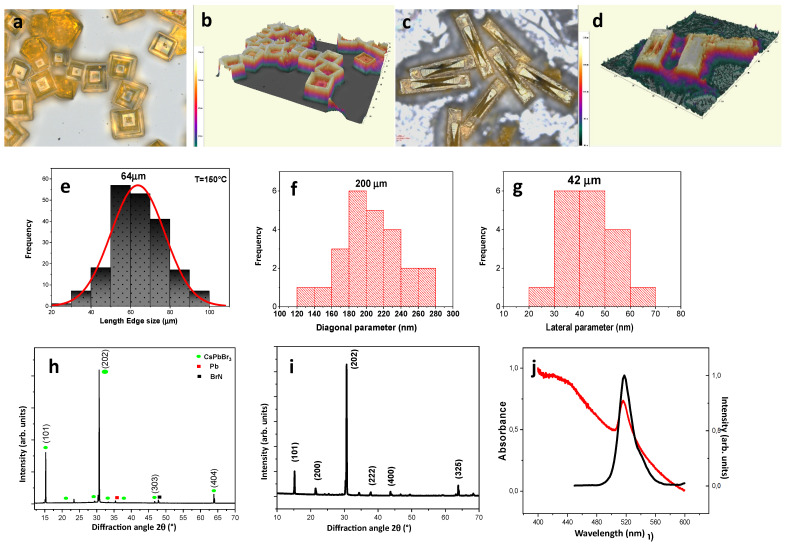
Structural characterization of CsPbBr_3_ microcrystals. (**a**) Optical micrograph (scale bar 28 µm) and (**b**) 3D optical micrograph image of some MCs. (**c**) Optical micrograph (scale bar 28 µm) and (**d**) 3D optical micrograph image of some MRs. (**e**) Histogram of the edge size parameter of MCs. (**f**,**g**) Histogram of the in-plane diagonal length (**f**,**g**) lateral size. (**h**,**i**) Typical XRD pattern of a MC and MR, respectively. (**j**) UV–visible absorption (red) and PL spectra (black) of an ensemble of MCs.

**Figure 2 materials-17-04043-f002:**
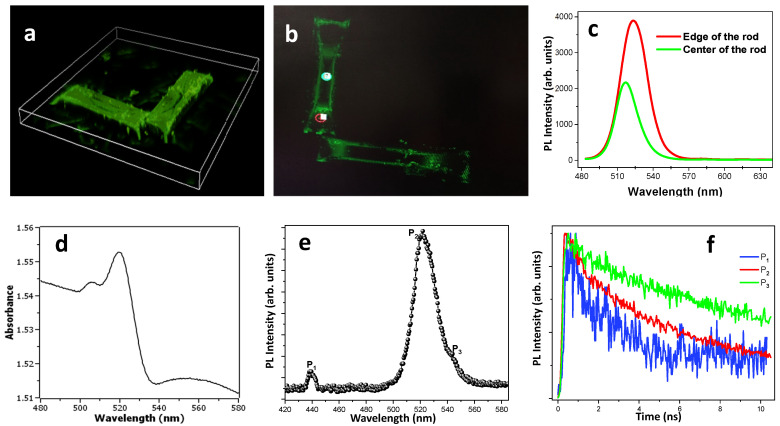
Optical measurements of the MRs. (**a**) 3D rendering and (**b**) top-view confocal microscopy image of two MRs. (**c**) PL spectra obtained from one MR at the spots marked in (**b**). (**d**) Absorbance spectrum, (**e**) steady state PL spectrum and (**f**) TRPL decay profiles for the three emission peaks P_1_ (blue), P_2_ (red), P_3_ (green), see text.

**Figure 3 materials-17-04043-f003:**
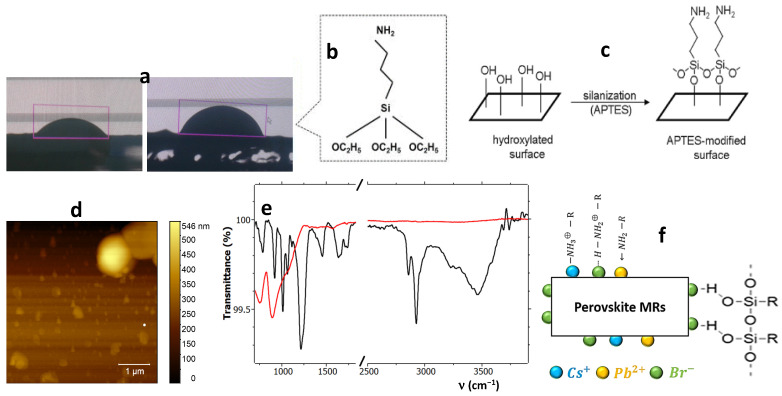
(**a**) CA measurements of (1) ITO-coated glass and (2) APTES/ITO-glass. (**b**) APTES chemical structure. (**c**) Silanization process of ITO-patterned glass substrate. (**d**) AFM image of APTES-modified ITO glass substrate. (**e**) FTIR spectrum of CsPbBr_3_ MRs deposited on APTES ITO-coated glass substrate. (black) and of MCs (red, substrate without APTES). (**f**) The schematic structure of the possible binding of the obtained rods with aminosilane layers.

**Figure 4 materials-17-04043-f004:**
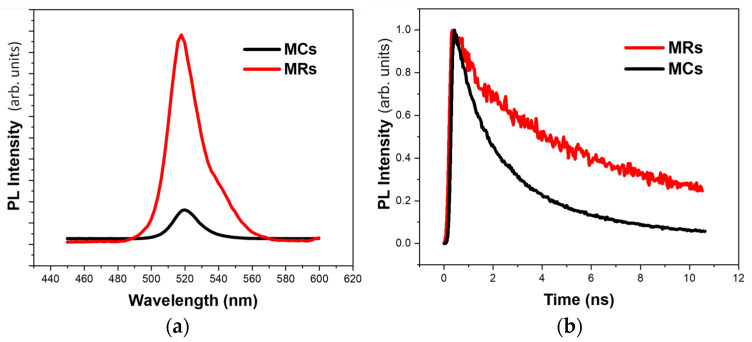
(**a**) PL spectra and (**b**) TRPL decay profiles of CsPbBr_3_ MCs (black line) and MRs (red line).

**Table 1 materials-17-04043-t001:** FTIR bands and peak assignments.

Wavenumber (cm^−1^)	Functional Group
3464	-OH
3258	-NH
2959–2924–2856	-CH
1715–1734	-CH
1638–1664	-NH_2_
1453	C-H
1244–1213	C-H
1012–1066	Si-O-Si
922	CH_3_ rocking of APTES
755–788	Ethoxy group of APTES

**Table 2 materials-17-04043-t002:** PL lifetimes of CsPbBr_3_ MCs and MRs.

Sample	*A* _1_	*τ*_1_ (ns)	*A* _2_	*τ*_2_ (ns)	*τ_Av_* (ns)
MCs	54.65	3.39	45.35	1.01	2.31
MRs	72.56	9.95	19.79	1.10	7.43

## Data Availability

The raw data supporting the conclusions of this article will be made available by the authors on request.

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
