# Peer review of "Single CsPbBr3 Perovskite Microcrystals: From Microcubes to Microrods with Improved Crystallinity and Green Emission"

_materials, 2024, doi:10.3390/ma17164043_

Round 1
Reviewer 1 Report
Comments and Suggestions for Authors
Abiedh et al. used a conventional drop casting technique to grow variously shaped CsPbBr3 perovskite microcrystals, such as microcubes and microrods, on flat substrates. They discovered that microrods could be cultivated on APTES-treated ITO glass. The study validated the APTES-ITO bond through contact angle measurements, FTIR, and PL spectroscopy. The microrods showed enhanced PL and longer lifetimes compared to microcubes, likely due to fewer defects. This research underscores the significance of substrate surface management in perovskite crystal growth and offers insights for optoelectronic device development. After addressing some major concerns, this manuscript can be considered acceptable.
The detailed concerns are as follows:
(1) For Section 2.2. Perovskite Preparation, the article mentions the use of a conventional drop casting method to grow microcrystals, but it does not provide detailed information on specific experimental conditions, such as solution concentration, the number of castings, annealing temperature, etc. It is recommended that the authors provide more detailed experimental parameters to facilitate the reproduction of the experiments by other researchers.
(2) From the XRD patterns of Figure 1g and Figure 1h, it can be known that MC has narrower FWHM and stronger crystallization than Mr. Why is the PL lifetimes of CsPbBr3 MCs in Table 2 shorter than that of MRs? The author should give a reasonable explanation.
(3) The author attributed the sudden jump at 320 nm to the edge characteristic attraction of APTES. However, according to the personal experimental test, the sudden jump signal was caused by the switching of the light source (halogen lamp switches to deuterium lamp) of the instrument during the test, not useful information. The author can give full consideration to this suggestion.
(4) Check and keep the font, size and other formats in all Figures consistent.
(5) In this paper, a possible mechanism is proposed to explain the anisotropic growth of nanorods, but there is a lack of in-depth theoretical analysis or computational simulation. If laboratory conditions permit, the author can consider whether to add theoretical calculation simulation data according to the situation.
Comments on the Quality of English Language1.It is suggested that "Due to the solution processability [1], low cost [2], high stability [3] and narrow emission line [4], all-inorganic cesium halide perovskites CsPbX3 (X=Cl, Br and I) have recently attracted high interest in optoelectronic applications [4,5].”
Revised as "Cesium halide perovskites CsPbX3 (X=Cl, Br, I), which are all-inorganic, have recently attracted high interest in optoelectronic applications due to their solution processability [1], low cost [2], high stability [3], and narrow emission line [4].”
2. It is suggested that “The excitation laser pulse width was 70 ps, the time resolution of the experimental setup was 40 ps.” be changed to “The excitation laser pulse width was 70 ps, and the time resolution of the experimental setup was 40 ps.”.
3.It is suggested that “For ideal silanization, all amino group of the APTES should be oriented away from the substrate surface [26].” be revised to “For ideal silanization, all amino groups in the APTES should be oriented away from the substrate surface [26].”.
Reviewer 2 Report
Comments and Suggestions for Authors
The authors have found that the shape of single crystals of CsPbBr3 can be controlled to be rod-like by using APTES.
Furthermore, they found that this geometry improves the PL lifetime.
This finding may lead to new developments in the field of optoelectronics.
I believe this paper needs minor revisions in order to be accepted by this journal. The points of concern are listed below.
1. In Figure 1f and Figure 3a, the numbering is further subdivided into 1, 2, and so on. This is very confusing, so I recommend using a, b, c,... and sequential alphabetical numbering.
2. In the FTIR in Figure 3e, why is only the MRs spectrum shown?
For comparison, the MCs spectrum is needed and a comparison discussion should be introduced.
3. In the MRs in Figure 4a, there appear to be two PL peaks, with a shoulder peak at 530~560 nm, which the author should mention and explain.
Round 2
Reviewer 1 Report
Comments and Suggestions for Authors
The author has effectively addressed significant concerns, and I recommend acceptance after minor revisions.
The authors need to address the following concerns:
1. Please check the subscripts of the chemical formula "CsPbBr3" in the title.
2. On page 4, the author mentions "A single drop of 40 mL volume." Should it be "40 µL volume" instead of "40 mL volume"?
3. It would be more aesthetically pleasing if the height of Figure 1b is consistent with that of Figures 1a and 1c.
4. The font of the x-axis titles in Figures 1g and 1i is incorrect; it should be 2θ. The font used in the figures looks odd.
5. The x-axis title in Figure 1g has a spelling error; it should be "wavelength," not "wavelengtht."
6. I recommend placing the labels for Figures 1f and 1g in the upper left corner inside the figures. Additionally, enlarging the figures will make the x- and y-axis titles clearer.
7. The x-axis title texts in Figures 2e and 2f are missing spaces, such as "wavelength(nm)" and "Time(ns)."
8. The lines in Figure 3e are too thin and difficult to see. I recommend making the lines bolder for better visibility.
9. I recommend reformatting Figure 1. Here is a suggested layout for the authors to consider:
